# Effect of Two-Port Laparoscopic Surgery on Pregnancy Outcomes of Patients with Concurrent Adnexal Masses

**DOI:** 10.3390/jcm11164697

**Published:** 2022-08-11

**Authors:** Ying-Xuan Li, Mu-En Ko, Ching Hsu, Kuan-Ju Huang, Bor-Ching Sheu, Wen-Chun Chang

**Affiliations:** 1Department of Obstetrics and Gynecology, National Taiwan University Hospital, National Taiwan University College of Medicine, Taipei 100, Taiwan; 2Graduate Institute of Clinical Medicine, National Taiwan University College of Medicine, Taipei 100, Taiwan

**Keywords:** two-port, laparoscopy, pregnancy, adnexal masses, delivery

## Abstract

Adnexal masses are common in pregnancy, with 2–10% of pregnancies presenting with an ovarian mass and approximately 1–6% of these masses being malignant. For suspected malignancy or masses with symptoms, surgery must be performed as early as possible. We retrospectively investigated the effect of two-port laparoscopic surgery on the outcomes of patients with concurrent adnexal masses between 2012 and 2019 (including large mucinous tumor, large teratoma, serous borderline tumor, and heterotopic pregnancy). Laparoscopic right partial oophorectomy was performed for a 27 cm ovarian mucinous tumor at a gestational age (GA) of 21 weeks, laparoscopic right oophorocystectomy for an 18 cm teratoma at a GA of 10 weeks, and laparoscopic left salpingo-oophorectomy for a 7 cm serous borderline tumor at a GA of 7 weeks after ultrasonographic confirmation of an intrauterine gestational sac with a fetal heartbeat. Laparoscopic excision of a tubal pregnancy was performed in a heterotopic pregnancy at a GA of 12 weeks with massive internal bleeding. Laparoscopic surgery is easier and safe to perform during early pregnancy because a smaller uterus allows for superior visualization. All of these patients had optimal postoperative recovery and normal spontaneous delivery at term. We discussed several aspects of treatment and delivery, namely treatment option (expectant management or surgery), surgery timing (early or advanced pregnancy), surgery type (laparoscopy or laparotomy), and delivery route (normal spontaneous delivery or cesarean section), in patients with concurrent adnexal tumors and their effects on pregnancy outcomes.

## 1. Introduction

Adnexal masses are common during pregnancy, with an incidence of 2–10%; approximately 1–6% of these tumors are malignant [1,2,3,4]. In patients with malignant tumors, surgery must be performed as early as possible. We investigated the effect of two-port laparoscopic surgery on the outcomes of patients with concurrent adnexal masses, including large mucinous tumors, large teratomas, serous borderline tumors, and heterotopic pregnancy.

We also discussed several aspects of treatment and delivery, namely treatment option (expectant management or surgery), surgery timing (early or advanced pregnancy), surgery type (laparoscopy or laparotomy), and delivery route (normal spontaneous delivery or cesarean section), in patients with concurrent adnexal tumors and their effects on pregnancy outcomes.

## 2. Materials and Methods

This retrospective study included four pregnant patients with concurrent adnexal mass who received two-port laparoscopic surgery at National Taiwan University Hospital between 2012 and 2019. This study received approval from the Research Ethics Committee of National Taiwan University Hospital (ID No. 202206021RIN). Patient characteristics, such as age, income, gestational age (GA), and side and size of the adnexal mass, were recorded. The operative findings and delivery outcomes, such as the operating time, estimated blood loss, hospital stay duration, delivery age, and birth body weight (BBW), were obtained by reviewing the medical records.

### Procedure

Two-port laparoscopic surgery has been described in previous studies [5,6,7]. Under general anesthesia, a 2 cm skin incision was made at the umbilicus. To prevent injury to the gravid uterus, the abdominal cavity entry was gained through the open technique (i.e., under direct vision) rather than through the blind Veress needle puncture. An XS Alexis wound retractor (Applied Medical, Rancho Santa Margarita, CA, USA) was placed through the umbilical wound, and a size-7 surgical glove was used to cover the wound retractor rim. A 10 mm trocar and a 5 mm trocar were inserted into the glove fingers; subsequently, an assistant 5 mm trocar was inserted into the left lower or upper abdomen on the basis of the gravid uterus size under laparoscopic inspection (Figure 1). Pneumoperitoneum pressure was set at 12 mmHg to avoid the delirious effects of CO_2_ on the fetus [8]. For ovarian cystectomy, the surface was incised and stripped completely without removing normal-appearing ovarian tissues. Bleeding from the remaining ovary was controlled by using bipolar electrocauterization. For the adnexectomy, the ovarian ligament, fallopian tube, and infundibulo-pelvic ligament of the target side were identified, dissected, and cut via LigaSure™ (Valleylab, Boulder, CO, USA). The specimen was then removed via Endobag (Endo pouch with suture tie, Unimax Medical Systems Inc., New Taipei City, Taiwan) from the umbilical wound. The abdominal wall was closed layer-by-layer, using 1-0 Vicryl. Finally, the skin was sealed with DERMABOND^®^ Mini Topical Skin Adhesive (Ethicon Inc., Raritan, NJ, USA). The operative time was defined as the interval between the initial skin incision to closure. Blood loss was calculated as the amount of aspirated fluid in the bottle.

## 3. Results

### 3.1. Laparoscopic and Pregnancy Outcomes of the Four Cases

Table 1 lists the laparoscopic and pregnancy outcomes of the four cases, presented in the order of GA. Neither surgical complications nor spontaneous abortions occurred. No preterm delivery, congenital defects, or neonatal complications were registered.

#### 3.1.1. Case 1: Seromucinous Borderline Tumor (GA of 8 Weeks)

This 32-year-old woman underwent laparoscopic right salpingo-oophorectomy for a serous borderline tumor sized 7 cm in 2005, at the age of 22 years (Figure 2A,B). No tumor recurrence was noted during a 5-year follow-up, and she gave birth to her first child in 2013. Three years later, in 2016, she was pregnant again, and the prenatal ultrasonography at a GA of 5 weeks revealed a 6 cm left ovarian cyst with sand-like content and multiple papillary components (Figure 2C). One intrauterine gestational sac (IUGS) was seen, but the fetal heartbeat (FHB) had not appeared yet. Laparoscopic left salpingo-oophorectomy (Figure 2D–F) was performed for a seromucinous borderline tumor at a GA of 8 weeks after the ultrasonographic confirmation of IUGS with FHB (Figure 2G). Oral progesterone at 100 mg, four times daily, and 8% vaginal progesterone gel (90 mg/tube) twice daily were administered till the GA reached 12 weeks (Figure 2H), when the placenta took over the function of progesterone production. She had a stable pregnancy and normal spontaneous delivery (NSD) in term. No tumor recurrence was noted during the regular follow-up performed at the outpatient department.

#### 3.1.2. Case 2: Teratoma (GA of 10 Weeks)

In 2015, this 34-year-old woman, G1P0, underwent a routine obstetric follow-up at a GA of 6 weeks, and an 18 cm ovarian cyst with compressing effects was noted. A laboratory examination revealed elevated levels of CA125 (631 u/mL). After the confirmation of IUGS and FHB, we performed laparoscopic right oophorocystectomy at a GA of 10 weeks by using a 10 mm trocar to punch directly into the tumor and suction out 1700 mL of fat-containing fluid from the tumor. The pathology was mature cystic teratoma, with the specimen measuring 17.5 × 14.0 × 5.0 cm in size and 475 g in weight. In 2016, she had an NSD at a GA of 38 weeks, with preeclampsia and intrauterine growth restriction; the BBW was 2232 g. One year later, in 2017, she developed preeclampsia during her second pregnancy but received an emergency Cesarean section (CS) due to abruptio placenta and intrauterine fetal death at a GA of 34 weeks. In 2018, Hashimoto’s thyroiditis was diagnosed and controlled by using Eltroxin 100 mcg once daily. In 2019, she was pregnant for the third time without preeclampsia, and a live male baby weighing 2352 g was delivered through CS at a GA of 38 weeks.

#### 3.1.3. Case 3: Heterotopic Pregnancy (GA 12 of Weeks)

This 33-year-old woman, G1P0, had a history of left ovarian teratoma sized 2 cm in 2017. During her regular prenatal examination in 2019, she had a sudden onset of abdominal pain at a GA of 5 weeks, with minimal ascites; a left ovarian torsion sized 5.4 cm was suspected (Figure 3A,B). Her symptoms were spontaneously relieved afterward. A follow-up ultrasonography at a GA of 11 weeks revealed early pregnancy with FHB and a suspected 7.3 cm left ovarian teratoma (Figure 3C,D). At a GA of 12 weeks, she had continuous abdominal cramping pain at the periumbilical area, with nausea and vomiting 2 or 3 times. The ultrasonography revealed a 9.9 × 6 cm left ovarian heterogeneous tumor with moderate ascites, besides the normal intrauterine pregnancy (Figure 3E,F). A laboratory examination revealed a decreasing hemoglobin level (10.7 g/dL to 8.1 g/dL to 6.5 g/dL). Because of suspected early pregnancy with left ovarian cyst rupture, emergency laparoscopic surgery was performed, which revealed a ruptured adnexal mass. The adnexal mass was resected, and the massive internal bleeding was stopped (Figure 3G–J). The pathology was a tubal pregnancy. She had an NSD at a GA of 37 weeks, with a BBW of 2954 g. One year later, in 2020, she was pregnant with a blighted ovum and received termination at a GA of 9 weeks.

#### 3.1.4. Case 4: Mucinous Tumor (GA of 21 Weeks)

This 33-year-old woman, G2P1, underwent laparoscopic bilateral oophorocystectomy in 2010. A right multilocular ovarian cyst sized 13.5 cm was noted at a GA of 5 weeks (Figure 4A,B). A follow-up ultrasonography at 12 weeks revealed an enlarged cyst sized 16 cm (Figure 4C,D). At a GA of 21 weeks, she reported compressing symptoms, with less food intake and shortness of breath when lying down; the ovarian tumor had enlarged to 27 cm (Figure 4E,F). The large gravid uterus posed difficulty in performing a total oophorectomy; therefore, a laparoscopic right partial oophorectomy after aspiration of 3000 mL mucinous fluid content was performed (Figure 4G,H) in October 2012. The pathology was mucinous cystadenoma. After surgery, intravenous ritodrine was administered for tocolysis, and the dose was gradually tapered. Neither uterine contraction in the nonstress test nor vaginal discharge was noted. She had an NSD at a GA of 39 weeks, with a BBW of 3102 g. Sixteen months later, in February 2014, a recurrent right ovarian cyst sized 7 cm was detected, and she received laparoscopic right salpingo-oophorectomy, with the operating time being 47 min. The pathology was mucinous cystadenoma again. The patient was followed up for 2 years, and regular menstruation and no tumor recurrence in the left ovary were noted.

## 4. Discussion

In our hospital, two-port operations have almost completely replaced three-port operations, including laparoscopic staging for the endometrial cancer. In the previous study [5], we demonstrated a short learning curve in two-port access for laparoscopic surgery for endometrial cancer using conventional laparoscopic instruments. The curves leveled off with ongoing maintenance of competence from the 11th and 16th cases for two operators.

### 4.1. Treatment Option: Expectant Management or Surgery

Most adnexal masses are diagnosed incidentally during the first-trimester ultrasound screening. Corpus luteum cysts are the most common type of adnexal masses, accounting for 13–17% of such masses. Corpus luteum cysts spontaneously regress in the second trimester [9,10]; therefore, close observation is a reasonable option. However, surgical management is required for persistent masses with suspicious malignancy. Up to 10% of persistent complex ovarian masses are ultimately diagnosed as malignant tumors [11]. Ultrasound of an adnexal mass revealing the characteristics of septations, solid components, or papillary components is suggestive of malignancy [12]. Other imaging modalities, such as computed tomography and magnetic resonance imaging, can be useful [13]. Tumor markers, such as CA-125, alpha-fetoprotein, beta-human chorionic gonadotropin, and lactate dehydrogenase, are of limited use because of their substantial alteration by pregnancy [14]. Risks associated with the expectant management of adnexal masses, such as benign cystic teratoma, serous cystadenoma, mucinous cystadenoma, and endometrioma, during pregnancy include rupture, torsion, emergent surgery requirement, labor obstruction, and malignancy progression. Because the uterus size increases in pregnancy, masses that persist throughout pregnancy are displaced upward out of the pelvic cavity. Occasionally, they may be incarcerated in the Douglas pouch throughout pregnancy, resulting in labor dystocia due to obstruction in fetal descent. However, except in heterotopic pregnancy, the risk of rupture and labor obstruction is less frequent (0–10%) in pregnancies with adnexal masses [15]. In patients with persistent or large ovarian masses and who are at a higher risk of an acute abdomen, surgical management is encouraged [16]. In this study, two cases had persistent growing ovarian masses causing compressing symptoms, one had ruptured heterotopic pregnancy, and one had suspected malignancy. 

### 4.2. Timing of Surgery: Early or Advanced Pregnancy

Most adnexal masses in pregnancy resolve spontaneously; therefore, expectant management in the first trimester is encouraged. However, in patients with malignancy suspicion (such as the borderline tumor in our Case 3) and complications (such as the torsion or rupture in our Case 3 with heterotopic pregnancy), an operation is suggested as early as possible. Laparoscopic surgery for adnexal masses during the first trimester is safe both for the mother and the fetus [17]. For example, in our infertility department, two cases of heterotopic pregnancy were diagnosed at the regular ultrasound follow-up after embryo transfer and underwent laparoscopic salpingectomy at a GA of 7 weeks and 8 weeks, respectively; both patients had a successful delivery at term. In patients with persistent large adnexal masses causing compressing symptoms, such as benign teratoma in our Case 2, surgical intervention is reasonable in the early second trimester for obtaining an acceptable operating field and allowing minimal uterine manipulation to prevent preterm contractions. Laparoscopic surgery at the end of the second or third trimester is more difficult to perform; for example, consider the laparoscopic total oophorectomy for the large mucinous tumor in our Case 4. In addition, incidental injuries to the gravid uterus caused by the Veress needle or trocar may result in bleeding, amniotic fluid leakage, and an adverse obstetric outcome [16]. 

### 4.3. Type of Surgery: Laparoscopy or Laparotomy

Laparoscopy performed by trained and experienced providers is safe during pregnancy [1,18,19,20,21]. Laparoscopy is a suitable alternative to laparotomy because of the advantages of shorter operating times and hospital stays [22,23]. Operative laparoscopy performed during early pregnancy is safe and feasible [16,17,24,25]. Laparoscopic surgery after the late second trimester of pregnancy poses technical challenges, including gravid uterus injuries caused by the Veress needle or trocar and difficulty achieving adequate visualization of the limited space between the adnexal mass and laparoscope in the umbilical trocar. To prevent gravid uterus injury, we used the laparoscopic entry technique with an open approach [26] rather than using a Veress needle. Moreover, our two-port method leaves the trocar outside the abdominal cavity (Figure 1A), with the laparoscope reaching the umbilicus (Figure 1B); this results in a greater distance between the adnexal mass and laparoscope, thereby providing a superior panoramic vision. Several studies have used the laparoscopic entry technique with the left upper quadrant approach to remove adnexal masses in the second trimester and have proved the technique to be feasible and safe [23,27,28]. 

Malignancy risk is one of the main concerns of an adnexal mass. The incidence of malignancy among persistent adnexal masses in pregnancy ranges from 1% to 6% [1,2,3,4]. The management and treatment of pregnant patients with ovarian cancer should consider GA at diagnosis, tumor histology, disease stage, obstetric complication risk, and patients’ preferences [29,30]. Because ultrasounds are conducted frequently during pregnancy, approximately 80% of pregnant patients with ovarian cancer are diagnosed with stage I disease [31]; therefore, fertility-sparing surgery may be a safe option [32,33]. Abdominal surgery can be performed either with laparotomy or laparoscopy. If surgical procedures are restricted during pregnancy because of the enlarged uterus and manipulation limitations, secondary surgery can be performed during a CS or after delivery [34]. Once ovarian cancer is suspected, surgery must not be delayed. If both ovaries are removed before a GA of 10 weeks, progesterone should be administered to replace the corpus luteum, as in our Case 1. The placenta is the primary provider of progesterone after 10 weeks of gestation. Therefore, surgery should be performed in the second trimester, if possible, to prevent the risk of spontaneous miscarriage. If the adnexal mass was noticed in the third trimester, waiting for fetal maturity may be recommended to avoid the risk of premature delivery due to surgical exploration. Corticosteroids for fetal lung maturation should be administered at least 48 h before surgery for a GA between 24 and 34 weeks [35]. Prophylactic perioperative tocolytic therapy might help delay preterm delivery [36]. Staging surgery can be performed, along with a CS, after the fetus is delivered at term [37]. 

### 4.4. Route of Delivery: Normal Spontaneous Delivery or Cesarean Section

In patients undergoing an antepartum operation for benign adnexal masses, the decision of delivery method is dependent on obstetric indications. If expectant adnexal masses incarcerate in the Douglas pouch, causing labor obstruction, a CS, along with tumor excision, may be performed. During a CS, any adnexal mass that is suspected of malignancy should be removed and sent for a frozen section. If a frozen section indicates malignancy, fertility-sparing surgery or staging surgery is performed after family counseling. In advanced ovarian cancer during pregnancy, when necessary, adjuvant chemotherapy or neoadjuvant chemotherapy should be administered in the second or third trimester and discontinued 3 to 4 weeks before delivery in order to prevent myelosuppression in the mother and neonate [38,39]. Planned laparotomy for CS and interval debulking surgery may be considered [30,34,40]. In this study, all cases had uneventful normal spontaneous delivery at term.

## 5. Conclusions

In patients with suspected adnexal mass malignancy, torsion, or rupture, an operation is suggested as early as possible. In patients with persistent large adnexal mass causing compressing symptoms, surgical intervention is reasonable in the early second trimester. Two-port laparoscopic surgery for the treatment of concurrent adnexal masses during pregnancy might be safe both for the mother and fetus. However, the three-port technique is still used worldwide, and the open technique, in the case of a large and/or malignant tumor, is a reasonable first-line option. Further research including a multicenter randomized control trial with standard three-port vs. two-port operations in the near future is required to support these conclusions.

## Figures and Tables

**Figure 1 jcm-11-04697-f001:**
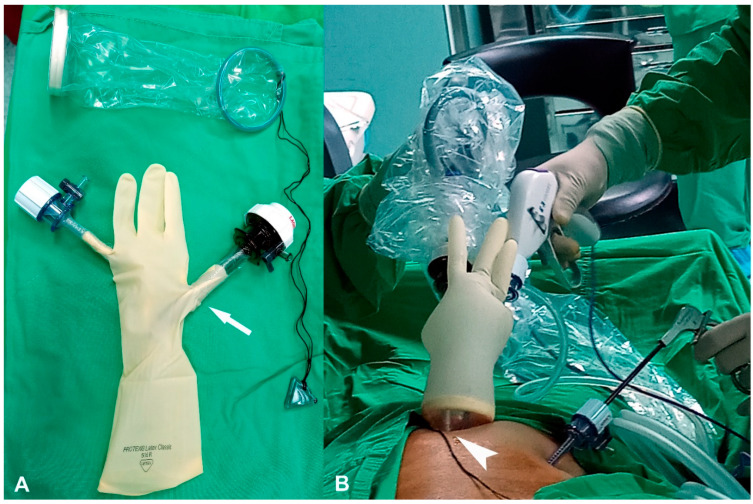
Two-port method of laparoscopic surgery. An XS Alexis wound retractor is placed through the umbilical wound, and the wound retractor rim is covered by a size-7 surgical glove. A 10 mm trocar and a 5 mm trocar are inserted into the glove fingers. Under laparoscopic inspection, an assistant 5 mm trocar is inserted into the left lower or upper abdomen on the basis of the gravid uterus size. (**A**) The white arrow indicates that the trocar is outside the abdominal cavity. (**B**) The white arrowhead indicates that the laparoscope level is as high as the umbilicus.

**Figure 2 jcm-11-04697-f002:**
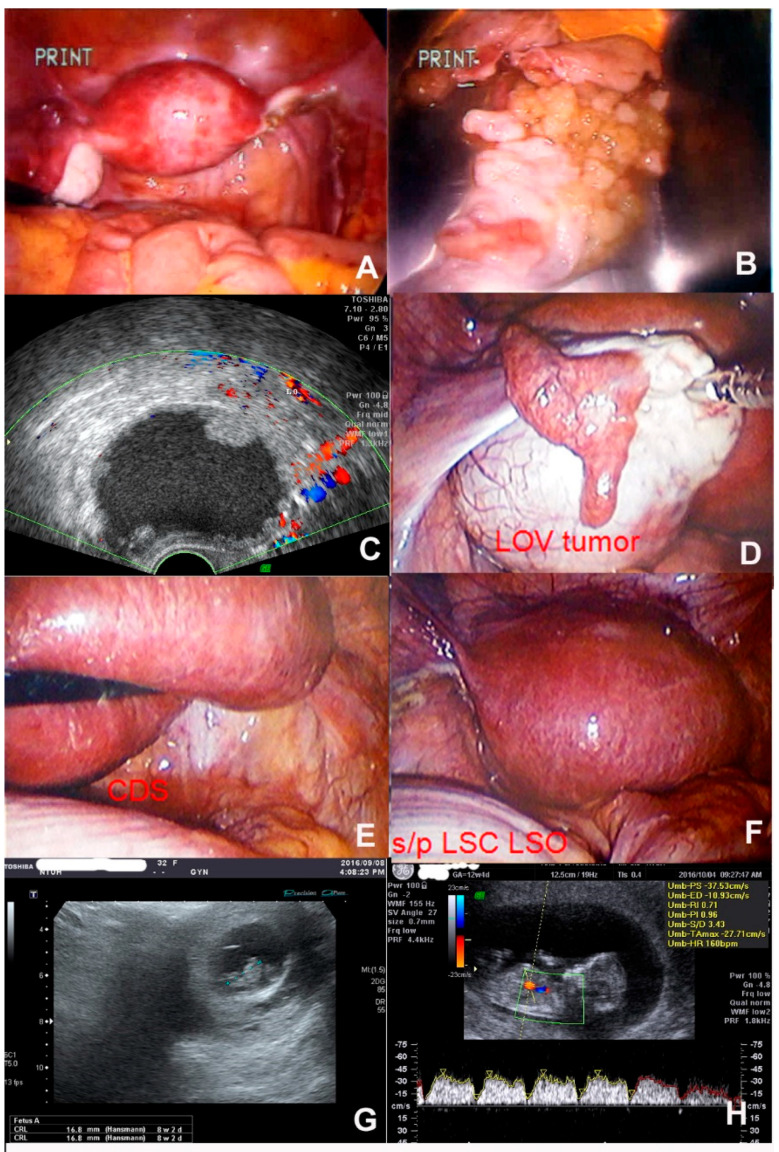
Seromucinous borderline tumor. (**A**) A 22-year-old woman who underwent laparoscopic right salpingo-oophorectomy for serous borderline tumor sized 7 cm; a normal left ovary. (**B**) Resected right ovary revealing a papillary solid growth inside. (**C**) Prenatal ultrasonography at GA 5 weeks revealing a 6 cm left ovarian cyst with sand-like content and multiple papillary components. (**D**) Laparoscopy revealing a left ovarian tumor without tumor outgrowth. (**E**) Soft gravid uterus that should be touched gently. No adhesion noted after the previous right salpingo-oophorectomy performed 10 years ago. (**F**) Wound and gravid uterus at GA 8 weeks after laparoscopic left salpingo-oophorectomy. (**G**) Ultrasonographic confirmation of an intrauterine gestational sac with fetal heartbeat at GA 8 weeks. (**H**) Ultrasonography revealing normal fetal growth at GA 12 weeks when the placenta took function. LOV—left ovary, LSC—laparoscopy, LSO—left salpingo-oophorectomy, CDS—cul de sac.

**Figure 3 jcm-11-04697-f003:**
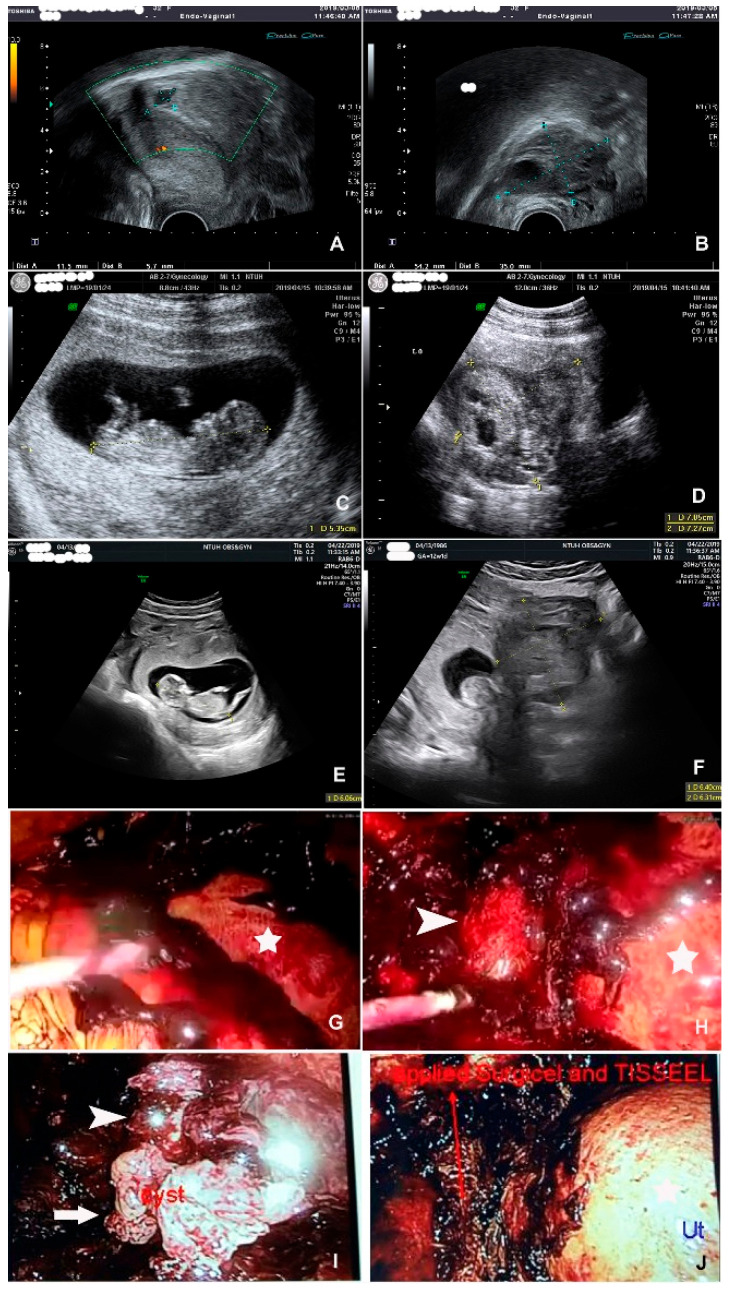
Heterotopic pregnancy. (**A**) Ultrasonography at GA 5 weeks revealing an intrauterine gestational sac. (**B**) Ultrasonography revealing a left ovarian mass sized 5.4 cm. (**C**) Ultrasonography at GA 11 weeks revealing early pregnancy with fetal heartbeat. (**D**) Ultrasonography revealing a left ovarian mass sized 7.3 cm, with a suspicion of teratoma. (**E**) Ultrasonography at GA 12 weeks revealing early pregnancy with fetal heartbeat and normal growth. (**F**) Ultrasonography revealing a left ovarian heterogeneous tumor with moderate ascites and normal intrauterine pregnancy. (**G**) Emergency laparoscopic surgery revealing massive internal bleeding covering the gravid uterus (white star). (**H**) Ruptured tubal mass after blood suction (white arrow head), the gravid uterus (white star). (**I**) White arrow indicates the left tubal fimbria end. (**J**). Surgicel and Tisseel application after operation and hemostasis to stop oozing.

**Figure 4 jcm-11-04697-f004:**
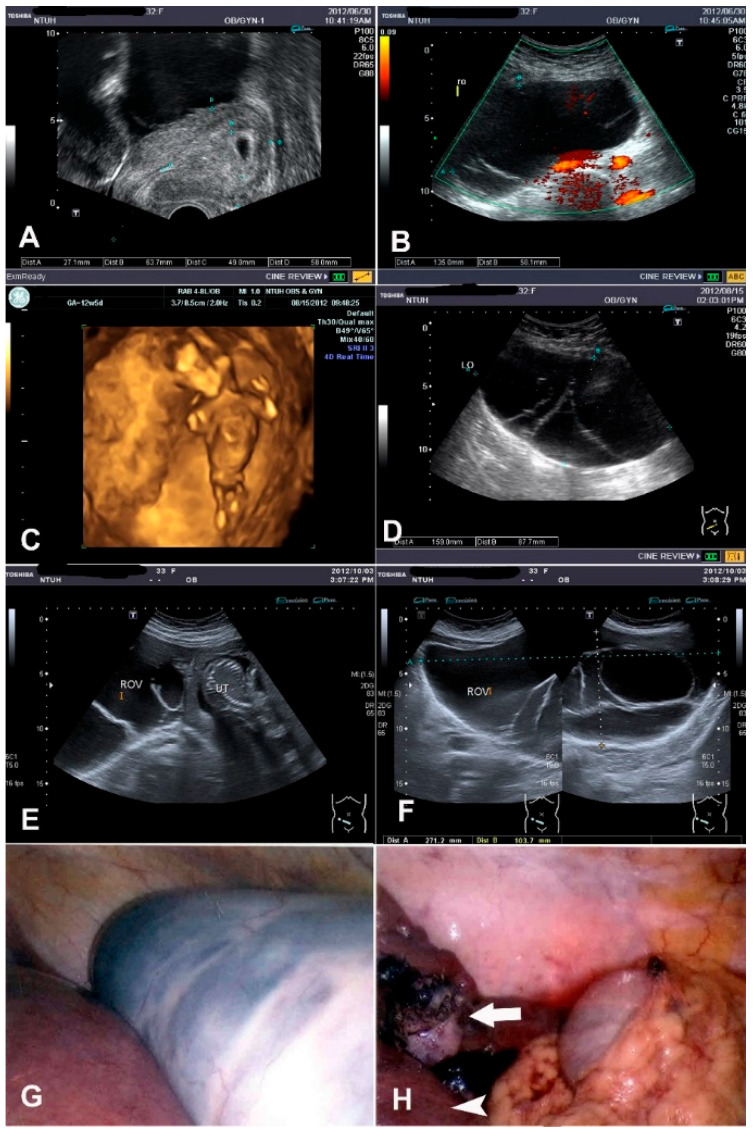
Mucinous tumor. (**A**) Ultrasonography revealing GA 5 weeks with an intrauterine gestational sac. (**B**) Ultrasonography revealing a right multilocular ovarian cyst sized 13.5 cm. (**C**) Three-dimensional ultrasonography at GA 12 weeks revealing early pregnancy with normal fetal growth. (**D**) Follow-up ultrasonography revealing a rapidly enlarging cyst (16 cm) at GA 12 weeks. (**E**) Ultrasonography revealing a right ovarian multilocular cyst and normal intrauterine pregnancy. (**F**) Ultrasonography revealing a rapidly enlarging cyst (27 cm) at GA 21 weeks. (**G**) Laparoscopic right partial oophorectomy after aspiration of 3000 mL of mucinous fluid content. (**H**) White arrow indicates the residual right ovary, which was difficult to remove totally because of the large gravid uterus (white arrow head).

**Table 1 jcm-11-04697-t001:** The laparoscopic and pregnancy outcomes of the four cases.

	Case 1	Case 2	Case 3	Case 4
	Borderline tumor	Teratoma	Heterotopic pregnancy	Mucinous tumor
Year	2016	2015	2019	2012
Age (y)	32	34	33	33
GP	G2P1	G1P0	G1P0	G2P1
GA (weeks)	8	10	12	21
Side	Left	Right	Left	Right
Size (cm)	7	17.5	6	27
operation	Salpingo-oophorectomy	Cystectomy	Excision	Partial oophorectomy
Operation time (minutes)	58	90	52	68
Hospital stay	2	5	4	4
Blood loss	10	50	50	10
delivery	NSD	NSD	NSD	NSD
DA (weeks)	38	38	37	39
BBW (g)	3008	2232	2954	3102

GP: gestation, partum. GA: gestational age. DA: delivery age. BBW: birth body weight. SO: salpingo-oophorectomy. NSD: normal spontaneous delivery.

## Data Availability

The study did not report any data.

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
