# Peer review of "Effect of Two-Port Laparoscopic Surgery on Pregnancy Outcomes of Patients with Concurrent Adnexal Masses"

_jcm, 2022, doi:10.3390/jcm11164697_

Round 1
Reviewer 1 Report
The abstract is very confusing, you don't understand much about how the study was conducted and the objective conclusions of the research.
It should be noted that this is retrospective study included 4 pregnant patients with concurrent adnexal mass who received two-port laparoscopic surgery at National Taiwan University Hospital between 2012 and 2019.
Please exclude the authors' opinions on the topic.
Abstract completely rewrite.
In the text, in M&M the technique of 2-port laparoscopic surgery is practically described (already extensively described in many important works already published).
The rest of the description of M&M was missing.
In the results, the case reports are commented out.
In the discussion, a mini review is made on the topic without properly discussing the results of the study, comparing them with the current literature.
In the conclusions the authors state (out of 4 case reports) that "the two-port laparoscopic surgery for the treatment of concurrent adnexal masses dur-278 ing pregnancy is safe both for the mother and fetus". Based on what? On what scientific result?
They concern the small number of patients on whom the study was conducted (only 4 case reports), on the type of study (retrospective), on how the paper was written, or on the lack of of any scientific clarity and editorial notion.
Author Response
Thank for the kind reviewers’ comments on this manuscript. We have made amendments according to the reviewers’ comments. Please see the attachment.

Reviewer 2 Report
(A) Overview/summary of the manuscript
This study evaluated the impact of two-port laparoscopic procedure on the results of surgical treatment of patients with adnexal masses during pregnancy.
The primary endpoint was the evaluation of the technique of two-port laparoscopic procedure during pregnancy. Secondary endpoint was assessment of impact of two-port laparoscopic surgery on pregnancy and evaluation of safety of this procedure during pregnancy.
Four pregnant women included to this retrospective cohort study. Study was conducted at National Taiwan University Hospital 43 between 2012 and 2019. All clinical data was collected from medical records. Results of each operated woman were presented as a clinical case.
In the results of study, authors evaluated the pregnancy safety after surgical treatment using two-port laparoscopic surgery.
In summary, authors recommended providing surgical treatment with adnexal masses as early as possible and in the second trimester.
(B) Introduction and discussion
• Relevant published work have been cited
• The authors highlighted the aims, significance and the novelty of the work
(C) Materials and methods
The methods and statistical analysis used are appropriate:
(D) Results
This manuscript offers original data. Subject matter manuscript has scientific interest and potential applicability. The conclusion section matches the structure of the Results and supported by the data presented.
The Discussion clearly written, justified by the results, and illustrate key points.
Results of this study do not conflict with the outcomes of previous studies.
Authors consider alternative interpretations and proffer practical implications.
(E) Quality of english language
· the level of the English is high
Remark: ethics not described (IRB approval)
Author Response
Thank for the kind reviewers’ comments on this manuscript. We have made amendments according to the reviewers’ comments as follows:
This study received approval from the Research Ethics Committee of National Taiwan University Hospital (ID NO. 202206021RIN).
Reviewer 3 Report
Please, check carefully the manuscript and make appropriate corrections.
Examples
Line 15: “pregnancy.” to “pregnancy).”
Line 16: “GA 21” to “gestational age (GA) 21”
Line 77: There is no caption for Table 1
Table 1: “OP time” to “OP time (minutes)”
Line 84: OP: operation
Line 176: There is no caption for Figure 4
Author Response

(The authors gave the same response as above.)

Reviewer 4 Report
This is another article of the authors on 2 port endoscopic technique designed in this situation to manage ovarian cysts and other adnexal masses in pregnancy.
In the 4 cases presented here the authors discuss advantages of using their novel approach and intend to convince readers to apply it in similar situations.
There are however stronger evidence lacking to support the change in laparoscopic procedure in pregnancy. 3 port technique is still world-wide used and in case of large and/or malignant tumor the open technique is a reasonable first-line option.
It would be valuable to add more arguments for authors' technique and of course to perform a multicentre RCT with standard 3 port vs 2 port operations in the near future.
Also the authors could provide the learning curve details and percentage of their technique to standard approach in their hospital.
Author Response

(The authors gave the same response as above.)
